# Nutrient-Level Evaluation of Meals Provided on the Government-Funded School Lunch Program in New Zealand

**DOI:** 10.3390/nu14235087

**Published:** 2022-11-30

**Authors:** Jamie de Seymour, Alessandro Stollenwerk Cavallaro, Laurie Wharemate-Keung, Sheryl Ching, Jasmin Jackson

**Affiliations:** 1Special Projects Team, Ministry of Education, Wellington 6011, New Zealand; 2College of Health, Massey University, Auckland 0632, New Zealand

**Keywords:** food insecurity, childhood poverty, evaluation, school lunch, nutrients

## Abstract

Approximately 1 in 6 children in New Zealand are living in households facing poverty and 14% of the population is food insecure. The Ka Ora, Ka Ako|Healthy School Lunches program aims to reduce food insecurity by providing access to a nutritious lunch every school day. This study analyzed the nutritional content of Ka Ora, Ka Ako meals and compared them to national and international standards. Meals were selected at random from approved menus. The suppliers covered by the 302 meals analyzed provide 161,699 students with a lunch (74.9% of students on the program). The meals were analyzed using Foodworks 10 nutrient analysis software. The nutrient content was compared against the New Zealand/Australia Nutrient Reference Values (NRVs) and to nutrient-level standards for international school lunch programs. A total of 77.5% of nutrients analyzed exceeded 30% of the recommended daily intakes. Protein, vitamin A and folate met the NRV targets and a majority of the international standards (55/57). Energy, calcium, and iron were low compared to NRVs and international standards (meeting 2/76 standards). Carbohydrates were low compared to international standards. The findings have been used to inform the development of revised nutrition standards for the program, which will be released in 2022.

## 1. Introduction

Approximately 1 in 6 children in New Zealand are living in households considered to be in poverty [1]. According to the 2017 UNICEF report card of 41 high-income countries, New Zealand had one of the highest rates of children living in a jobless household (one in seven) [2]. In New Zealand’s most disadvantaged communities, as many as 40% of parents indicated that they ran out of food ‘sometimes’ or ‘often’ and around one in five children in New Zealand live in households that struggle to put enough good-quality food on the table [3]. Food security for families in New Zealand is a growing concern, particularly given the recent large rise in inflation. In 2018, the United Nations, Food and Agriculture Organization, and the World Health Organization reported that approximately 14% of the New Zealand population is food insecure [4]. The New Zealand population is made up of approximately 70.2% European, 16.5% Māori, 8.1% Pacific peoples, 15.1% Asian, and 2.7% other ethnicities [5]. Data from national health surveys show that the prevalence of food insecurity is much higher in Māori and Pacific populations in New Zealand. A recent New Zealand Health survey found that 37.1% of Pacific children and 28.6% of Māori children lived in food-insecure households [6]. In the 2019 December quarter, 307,291 grants for food were provided at a cost of $30.3 million NZD, more than triple the number in 2014 [7]. This number has grown substantially since the start of the COVID-19 pandemic. In the June 2020 quarter, 566,647 Special Needs Grants (totaling over $64 million) for food were approved by the New Zealand Ministry of Social Development [8].

Food insecurity impacts not just the quantity of food provided for New Zealand children, but also the variety of food, which in turn affects the range of nutrients available. Children in food insecure households in New Zealand are significantly more likely to:»not meet fruit and vegetable dietary guidelines;»not eat breakfast and/or lunch;»eat fast food more often;»eat processed snacks and fizzy drinks more often [6,9].

In Canada, adolescents living in food insecure households had higher levels of nutrient inadequacies, particularly in relation to protein, vitamin A, thiamin, riboflavin, vitamin B6, folate, vitamin B12, magnesium, phosphorus, and zinc [10]. Nutrient intake plays an important role in general health and wellbeing, as well as brain development and in turn cognitive function, which can affect school achievement. Nutrition in early years can influence brain structure and development [11]. Nutrition is also important for brain function, as nutrients are required for the construction of neurotransmitters [11]. Intake/status of iodine, iron, omega-3 fatty acids, zinc, vitamin B12, vitamin B6, and folate have been associated with cognitive function and academic abilities [11,12].

Food insecurity has been linked to poor school attendance, engagement, and academic achievement. In the United States of America, children between 6–11 years of age facing food insecurity had significantly lower arithmetic scores, were more likely to need to repeat a year level at school, and had social and behavioral difficulties [13]. In the same study, food insecure teenagers also had a higher rate of social and behavioral issues and were more likely to have been suspended from school [13]. Similarly, in Canada it was found that food insecurity was associated with lower standardized scores on reading and mathematics exams sat by 10–11 year olds [14]. A systematic review of 21 studies conducted in developed nations (United States, Canada, United Kingdom, and Australia) found that food insecurity in children, even at marginal levels, was associated with poorer school attendance, achievement, interpersonal/behavioral skills, and increased mental health concerns such as anxiety and depression, hyperactivity and inattention [15].

The Ka Ora, Ka Ako|Healthy School Lunches program was piloted in a selection of New Zealand primary schools in 2020, with the aim of reducing food insecurity by providing access to a nutritious lunch every school day. In May 2020, a major expansion of the lunch program was announced as part of the COVID-19 Response and Recovery scheme to help cushion the blow of COVID-19 impacts on students living in already socio-economically disadvantaged households. The program was expanded from approximately 21,000 to approximately 220,000 students, including secondary school students. Schools are eligible for inclusion into the program based on a cut-off on the New Zealand Ministry of Education’s Equity Index [16]. The program uses a universal model—a free lunch is provided to all students within eligible schools who join the program. As a result, approximately 25% of Year 1–13 students across New Zealand are receiving a free school lunch.

Nutrition guidelines were introduced into the program in June 2020 [17]. The nutrition guidelines for the program were based on the New Zealand Ministry of Health’s guidance for food and drink in schools [18]. This guidance is food-based, using a traffic light system which classifies food items as red, amber, or green. Meals on the program should consist of mostly green category ingredients; amber items are limited to 25% or less of the Ka Ora, Ka Ako meals; and red items are not permitted on the Ka Ora, Ka Ako program. In addition to meeting the Ministry of Health traffic light criteria, the Ministry of Education also has minimum criteria that suppliers must meet regarding meal size (grams), protein-containing foods and vegetable content of the meal (Table 1). Improving nutritional status of food insecure children in New Zealand is one of the pathways expected to lead to improvements in wellbeing, learning and achievement. Studies have highlighted that improvements in attendance, learning and achievement may not be observed until two or more years after lunch program initiation (or even five years, as one study found [19]) [20]. Therefore, evaluating the nutritional contribution of the lunch meals is important for assessing whether the lunch program is providing adequate nutrition to meet the needs of children on the program and thereby ensuring the program has the potential to realize benefits in wellbeing, learning and achievement.

Many school lunch programs internationally have nutritional guidelines or standards to encourage (or enforce) suppliers to produce a nutritious school lunch. The guidelines are varied between countries and range from nutrient-level standards/targets, food-based guidelines, and some are a combination of both. A number of studies have investigated nutrient provision from school lunch programs and compared their findings to the country-specific dietary recommendations or lunch-program nutrient standards. Most studies have predominantly investigated the macronutrients (protein, fat, carbohydrates), energy, and fiber in the school lunch meals, with a select few also analyzing a more thorough range of vitamins and minerals. Overall, there was a trend in the literature showing that meals provided on the lunch programs internationally tended to not meet the country-specific standards for carbohydrates (below the country-specific standards in Thailand, Slovenia, Sweden, China, Ghana, Portugal, and Iceland), energy (below standards in Thailand, Ghana, Slovenia, Finland, China, Iceland, Portugal, and Sweden), and fiber (below the country-specific standards in USA, Sweden, Finland, and Portugal) [21,22,23,24,25,26,27]. These studies emphasize the importance of conducting nutrition-focused evaluations of school lunch programs, as even in countries with long-standing programs and nutrient standards, there are still clear areas for improvement. The findings from our study add to this body of research.

The aim of this study was to evaluate the nutritional content of Ka Ora, Ka Ako meals from approved menus on the program. The objectives of the study were to:(1)determine the contribution of Ka Ora, Ka Ako lunch meals to the daily nutrient requirements of students (assessed against New Zealand/Australia Nutrient Reference Values [28])(2)compare the nutritional value of the Ka Ora, Ka Ako meals to international lunch program nutrient standards.

## 2. Materials and Methods

### 2.1. Meal Selection and Analysis

Each school term (in New Zealand there are four school terms in a year ranging from 8–10 weeks), as part of their contractual agreement, lunch providers on the program submit their menus to the Ka Ora, Ka Ako nutrition team at the Ministry of Education. Menus are approved if they meet the criteria set out in the nutrition guidance [17]. For this study, meals were selected at random from lunch providers’ approved standard menus between Term 3, 2021 and Term 2, 2022. Meals from special dietary menus were not included in the analysis. Meals were selected from each of the three year groups served: years 0–3 (age 5–8), years 4–8 (age 9–12), and years 9+ (age 13–18). Suppliers were selected using a random number generator on the list of suppliers for each participating school. Once all the data were entered, they were checked against the list of suppliers for each year group. Meals were added or removed where necessary to be representative and ensure that meals from each of the top ten suppliers were included, as they cover a large number of schools (473 (49.5%) schools) and students (133,231 (60.9%) students) participating in Ka Ora, Ka Ako. The data was then quality checked and peer reviewed before being analysed.

A total of 302 meals were included in the analysis. The years 0–3 meals analysed were from 50 suppliers; years 4–8 meals were from 44 suppliers; and years 9+ meals were from 58 suppliers. The suppliers covered by the 302 meals included in this analysis provide 161,699 students with a Ka Ora, Ka Ako lunch on a given day (74.9% of students on the program). The breakdown of the meals included in the analysis can be found in Table 2. Included in the analysis are meals from both suppliers (external model) and schools preparing their own lunches (internal model). The external model is where a school is provided lunches by a food business (or other non-commercial entity). External suppliers usually operate a central kitchen and deliver lunches to one or more schools daily. Occasionally, an external supplier will set up a kitchen on school grounds, however the lunch supplier still operates as a business and manages the end-to-end lunch provision process. The internal model is where the school takes on the responsibility of making their own lunches within the school, rather than outsourcing this to an external supplier.

The nutritional content of the 302 Ka Ora, Ka Ako meals was analysed using Foodworks 10 nutrient analysis software. New Zealand FOODfiles 2016, Aus Foods 2019, Ausbrands 2019, and Australian Food Composition Database were the food databases used in Foodworks 10 to collect nutrient information. Medians were calculated for 21 nutrients in the lunch meals and reported with their corresponding 25th and 75th percentiles. Nutrient levels were compared between meals supplied by internal-provision schools and external providers using the Mann–Whitney U test. *p*-values < 0.05 were considered statistically significant.

### 2.2. Comparison of Meals to National and International Standards

As there are no nutrient-level school lunch program standards in New Zealand, the nutrient contributions from Ka Ora, Ka Ako meals were compared against the New Zealand/Australia Nutrient Reference Values (NRVs) [28]. For most of the micronutrients (vitamins and minerals) considered in this analysis, as well as energy and protein, the levels in the meals were compared to the corresponding recommended dietary intake (RDI). In the case of fiber, where no RDI exists, the levels in the meals were compared to the Adequate Intakes (AI). For the purpose of this evaluation, we used a target contribution from the program as one third of recommended daily intakes for the nutrients analysed (with values above 30% deemed acceptable); this was similar to cut-offs used across a number of countries, including Japan (ranged from 33–50% depending on the nutrient), Taiwan (ranged from 33–40%), and South Korea (33% for all selected nutrients) [29]. However, it should be highlighted that consuming a third of daily energy needs within one meal is not always feasible, particularly for children. Therefore, comparisons of energy provision to a third of daily requirements should be interpreted with caution. Within each year group (0–3, 4–8, and 9+) the nutrient RDIs were calculated for a female and male in the lower and upper age bracket ie a 5-year-old and 8-year-old for years 0–3. Although adequate sodium intake is important for physiological processes, in New Zealand, high sodium intakes are a public health concern [30]. Therefore, sodium levels in the meals were compared to Upper Levels (UL) in the NRVs, to check the meals were not providing sodium in levels that were likely to be detrimental to health. Carbohydrate levels as a percentage of energy (%E) were compared to the Acceptable Macronutrient Distribution Ranges (AMDR) in the NRVs. The AMDR for carbohydrate is 45–65%E. It is recommended in the NRVs that saturated fat and trans-fat combined contribute a limit of 8–10%E, therefore 10%E was used as the cut-off for saturated fat contribution from the meals. Reference values from the NRVs were used when selecting height and weight brackets for the different age groups analyzed.

The levels of the key nutrients provided in the Ka Ora, Ka Ako meals were then compared to published nutrient-level standards for international school lunch programs (Appendix A). Not all international school lunch programs have publicly available nutrient-level standards, but many have been documented in the literature. In some instances, these are mandated for the program, and in others they serve as recommendations for suppliers.

## 3. Results

### 3.1. Comparison of Ka Ora, Ka Ako Meals to Nutrient Reference Values for Australia and New Zealand (NRVs)

The median levels of each nutrient in the meals provided to each year group and the corresponding 25th and 75th percentiles are displayed in Table 3.

Table 4 shows the percentage contribution to daily requirements for each of the nutrients in the meals. Appendix A shows the percentage of meals analyzed in each category which exceeded a third of the RDIs (or AI for fiber). Of the 204 nutrient comparisons to the NRVs in Table 4, 158 (77.5%) exceeded 30% of the RDIs (AI for fiber). Energy, calcium, and iodine were below 30% across all age/sex categories.

In the years 0–3 meals, with the exception of energy, calcium, iron, and iodine, all of the 13 remaining nutrients exceeded a third of daily requirements target across all age and sex groups.

In the years 4–8 meals, riboflavin, vitamin B6, magnesium, and selenium were marginally below a third (≥30%). Fiber was also marginally below the target (32.0%) for males. The remaining 10 nutrients exceeded a third of daily requirements across all age and sex categories.

In years 9+ meals, riboflavin was marginally below the target in 18-year-old females (31.8%) but significantly below the target in 18-year-old males (26.9%). Fiber was marginally below the target in 18-year-old males (32.2%), as was vitamin B6 (30.8%) and selenium (32.1%). Magnesium and iron were significantly below the target in both 18-year-old males and females; with iron having the lowest contribution to daily requirements for 18-year-old females (21.9%). The remaining seven nutrients exceeded a third of daily requirements across all age and sex categories.

Sodium levels in years 0–3 meals reached 35.8% of the UL; levels in years 4–8 meals reached 29.8% of the UL; and levels in the year 9+ meals reached 35.9% of the UL.

Carbohydrate levels in years 0–3 meals contributed 43.6%E; carbohydrate levels in years 4–8 meals contributed 47.0%E; and carbohydrate levels in years 9+ meals contributed 41.8%E.

Saturated fat levels in years 0–3 meals contributed 10.5%E; saturated fat levels in years 4–8 meals contributed 10.2%E; saturated fat levels in years 9+ meals contributed 10.1%E.

### 3.2. Comparison of Nutrient Content of Meals from Internal-Provision Schools and External Providers

Analyses comparing the nutrient levels (and meal size) of the meals from internal-provision schools and external providers showed significant differences between meal size and seven nutrients in the years 0–3 meals; meal size and two nutrients in the years 4–8 meals; and meal size and six nutrients in the years 9+ meals (Table 5). All significant nutrients (and meal size) were in higher amounts in the internal meals when compared to the meals from external providers. Only nutrients with statistically significant differences have been included in Table 5.

### 3.3. Comparison of Nutrient Content of Ka Ora, Ka Ako Meals to International Lunch Program Standards

The tables comparing the average nutrient contributions from Ka Ora, Ka Ako meals to international standards are displayed in Appendix A.

Protein (grams), vitamin A and folate levels, which met the NRV targets, also met a majority of the international standards (24/26 standards, 21/21 standards, and 10/10, respectively). However, due to the low amount of carbohydrates in the meals on Ka Ora, Ka Ako, protein as a %E exceeded all eight suggested %E ranges in the international standards.

Energy, calcium, and iron, which were low when compared to NRVs, were also low compared to international standards, meeting only two of the 76 standards across the three nutrients.

Interestingly, thiamin, riboflavin, and vitamin C, which were not of concern compared to NRVs, were found to be below many of the international standards (meeting only 4/11, 1/10 and 13/23 standards, respectively).

Although saturated fat in grams was below the maximum amounts recommended in 10/11 standards, saturated fat was above all of the 14 standards for %E. Carbohydrates were below all of the 11 standards for grams and all 11 standards for %E.

Sodium levels were below the recommended maximums in 9 of the 19 standards, with the years 9+ meals being least likely to pass the standards (passed 1 of the 6 standards).

## 4. Discussion

This study provided a comprehensive analysis of the nutritional quality of a selection of 302 meals from Ka Ora, Ka Ako, the government-funded school lunch program in New Zealand. This is the first nutrition-based evaluation of the program. The nutrition standards on Ka Ora, Ka Ako are food-based. Food-based nutrition standards are the simplest and most practical way to ensure suppliers can comply with guidance to provide healthy school lunches. This study allowed was an opportunity to check how the nutrient contribution of meals from supplier menus, approved according to the food-based Ka Ora, Ka Ako nutrition guidelines, compare to children’s daily requirements and how they align with international nutrient-level benchmarks. Our findings highlighted that the analyzed meals from Ka Ora, Ka Ako menus were nutrient dense and made of high-quality, nutritious foods to support children’s growth, development, wellbeing, and learning. Overall, 77.5% of the analyzed nutrients exceeded 30% of the RDIs (AI for fiber) and 66% were above a third of children’s daily needs. Two nutrients, protein and niacin, were provided in levels which exceeded 100% of the daily needs of students in some age groups and five nutrients were consistently below 30% of the RDIs or were low by international standards: energy, carbohydrates, iron, calcium, and iodine. The findings from this study have been used to inform improvements in the revised nutrition standards for the program, which are due to be released in the final quarter of 2022.

The nutrients which surpassed a third of daily requirements in analyzed lunch meals across all categories were thiamin, niacin, vitamin B12, vitamin A, and folate. Vitamin B12 and folate status in particular have been associated with school attendance, cognition, and academic achievement; these nutrients meeting a third of daily requirements is a strength of the program. A longitudinal study in Colombia of 3156 children aged 5–12 years found that vitamin B12 deficiency was associated with a significantly increased risk of needing to repeat a grade (2.36 fold) and increased school absences (1.89 fold) [31]. In the USA, a study of 2014 children 6–16 years of age from the NHANES-III study found that higher vitamin B12 status was associated with higher math and digit span test results, which are measures of attention and short-term memory [32]. Nguyen et al. (2013) also found that even after adjusting for confounding variables such as sex, age, race/ethnicity, and income-poverty ratio, children with the highest folate intake scored higher on reading and mean block design (assesses perceptional reasoning index and executive function) [32]. In Sweden, a study of 358 15-year-olds also found that folate intake was significantly associated with academic achievement (sum of school grades) and that this association remained significant even after adjustment for other predictors [33]. Thiamin deficiency, although not directly linked to school achievement, has been found to be associated with increased aggression, irritability, and anti-social behavior, particularly in adolescents, which could impact the learning environment in a classroom [34]. Meeting the target for these nutrients will help increase the probability of the program achieving wellbeing, learning, attendance, and engagement outcomes.

Protein levels exceeding 100% of NRVs in some age groups was not deemed an area of concern as there is little evidence to indicate adverse effects of protein intake at this level when consumed from foods, unless it is displacing other important nutrients [35]. It is also worth highlighting that common breakfast meals, such as cereals and toast are often low in protein, as are foods typically consumed during morning tea (such as crisps and snacks, fruit, and biscuits) [36]. Therefore, it would be beneficial for the lunch meal to provide more than 50% of daily requirements. Protein-rich foods are also some of the most expensive everyday foods and therefore, are likely to be more limited in households experiencing food insecurity. Ka Ora, Ka Ako meals providing 50% or more of daily requirements is a strength of the program. The protein levels provided in the Ka Ora, Ka Ako meals were not high when compared to what has been reported in studies of school lunch program provision internationally. In particular, the protein levels of the years 4–8 meals in Ka Ora, Ka Ako were lower than those reported in studies of lunch programs in China [21], Slovenia [25], Sweden [26], Finland [26], Iceland [26], and Portugal [27].

Niacin was also found to be in levels exceeding 100% of daily needs in some age groups. The upper level of intake (UL) in the NRVs for niacin intake is focused on intake from fortified foods or supplements. With chicken, cheese, and beef contributing more than 50% of niacin in the Ka Ora, Ka Ako meals (not niacin-fortified foods), the contribution of the lunch meals towards the UL is not likely to pose any health risk to students on the program. As such, no recommendations have been made to reduce niacin content of meals on the program.

Energy levels in the analyzed Ka Ora, Ka Ako meals were low in all age groups, compared to the NRVs and to international standards. Providing a third of children’s daily energy needs in a single meal is a high standard and not overly realistic for a child, from a consumption perspective. The New Zealand Food and Nutrition Guidelines for Healthy Children and Young People state that children should have three meals and two or three snacks per day to achieve their energy requirements [37]. The Ka Ora, Ka Ako meals contained less than a third of daily energy needs but most other nutrients analyzed were provided in levels which exceeded a third of daily needs, indicating that the meals on the program were very nutrient-rich, low-kilojoule meals. A large number of the international studies of school lunch provision have also found that energy provision was low compared to their country specific standards/recommendations [21,25,26,27].

Relative to international lunch program nutrient standards, the carbohydrate content of the analyzed Ka Ora, Ka Ako meals was very low. This is an aspect of the program that should be addressed. Interestingly, other studies also found that carbohydrate provision on school lunch programs was below their country-specific standards/recommendations (Slovenia [25], Sweden [26], Iceland [26], and Portugal [27]). An easy solution to increase the amount of carbohydrates provided on Ka Ora, Ka Ako is to add a minimum requirement for carbohydrate-rich foods into the nutrition requirements, much like the minimum requirement for protein-rich foods. Increasing carbohydrates will in turn increase energy content of the meals, providing a solution to two of the key areas for improvement based on these findings. The fiber content of the analyzed meals was sufficient (passing 12 of the 15 international standards), reinforcing that the carbohydrates provided on the program were predominantly nutrient-rich wholemeal/wholegrain options, which is in line with the program’s current nutrition requirements. The current nutrition guidance on the program encourages almost exclusive use of wholemeal grain products including brown rice, wholemeal pasta, high-fiber, low sodium wholegrain breads. As fiber can increase satiety [38] it is important to allow the inclusion of lower fiber options when increasing the carbohydrates in Ka Ora, Ka Ako meals. A meal too high in fiber could have children feeling satiated without finishing their meal and therefore not reaping the benefits of all the nutrients included in that meal.

Iron was below 30% of the RDI in half of the age/sex groups analyzed. With inadequate iron intake in 43.9% of menstruating females in the last New Zealand Child Nutrition Survey, this is a nutrient of concern [39]. Iron status is associated with cognition and academic performance. A systematic review published in 2022 assessed the findings from 50 studies (26 observational and 24 intervention studies) investigating iron status or iron-containing interventions and their association with cognition and academic performance in adolescents [40]. Samson et al. (2022) concluded that iron status may be linked to academic performance and that iron supplementation could improve performance, attention, and concentration.

One option to increase iron provision in Ka Ora, Ka Ako meals is to include a recommendation to suppliers that red meat (ie roast beef/mince dish) is provided in at least one lunch meal per week. The financial implications of this suggestion would need to be considered before implementation, as the cost of red meat in New Zealand is much higher than alternative protein options such as chicken. Most studies of lunch programs internationally found that iron provision was sufficient, with only studies in Ghana [24], Finland [26], Sweden [26] and Iceland [26] reporting iron provision below their country’s standards/recommendations. The iron provided in meals from Sweden and Finland was below the levels provided in Ka Ora, Ka Ako meals for students in a similar age bracket.

Calcium levels in the analyzed Ka Ora, Ka Ako meals were below 30% of the RDIs in all age/sex categories and only passed two of the 22 international standards. This is particularly concerning given that the New Zealand Child Nutrition Survey findings indicated that close to one in five New Zealand females in the study had inadequate calcium intake [39] and the Milk in Schools program, which previously provided milk in many primary schools across New Zealand, ceased in 2021. Calcium needs of the most vulnerable and food insecure children in New Zealand should be a priority. Many international school lunch programs include milk as a drink (including the USA, Finland, Sweden, and England [41,42,43]). However, there is currently insufficient funding in Ka Ora, Ka Ako to mandate milk provision as a drink on the program. This is an area of investment that should be considered either within or outside of Ka Ora, Ka Ako to address the low calcium intakes of New Zealand children.

Iodine levels in the analyzed Ka Ora, Ka Ako meals were below 30% of the RDIs in all age/sex categories. Since the mandated fortification of most breads in New Zealand with iodized salt in 2009, modelled data showed that children’s iodine intake had increased significantly since the low levels reported in the last New Zealand Children’s Nutrition Survey [44]. A study of iodine levels in New Zealand school children in 2015 (post-fortification) also showed adequate iodine status [45]. Given the regular use of wholemeal bread products on the program, a low level of iodine in Ka Ora, Ka Ako meals was not anticipated. Very few studies have looked at iodine contributions from school lunch programs. Martins et al. (2020) is one of the few studies to assess iodine contribution from a school lunch program in Portugal [27]. They found that none of the 448 meals analyzed met their recommendation of 36 µg of iodine from the lunch meal (30% of recommended daily intake).

Saturated fat as a percentage of energy was marginally higher than the NRV recommendations (less than 8 to 10%E of trans fats and saturated fats combined), ranging from 10.1% to 10.5%. However, the gram amounts of saturated fat in the analyzed Ka Ora, Ka Ako meals were not of concern by international standards (under 10 of the 11 recommendations for maximum saturated fat). Therefore, there is no need to reduce the amount of saturated fat in the Ka Ora, Ka Ako meals. As detailed previously, the analyzed Ka Ora, Ka Ako meals were low in energy and carbohydrates; increasing carbohydrates in future meals will reduce the percentage contribution of saturated fat to below 10% energy. This is a simple and easily achievable solution to the marginally high saturated fat (as a percentage of energy).

Sodium levels were high in the meals compared to NRV ULs and international standards (only meeting 9 of the 19 standards). The main contributor of sodium on Ka Ora, Ka Ako was wholemeal bread products (Appendix A). A reduction in wholemeal bread products on the program is not recommended as they contribute significantly to the provision of other important nutrients, including iodine and carbohydrates which are already low in the meals. Nutrition standards on Ka Ora, Ka Ako include sodium limits for ‘green’ breads, and it is not practical to tighten these without heavily limiting the options for suppliers. Two of the other main sodium sources which could be reduced without having a negative impact on the other nutrients on the program are sauces and processed meat. Therefore, it is recommended that there is a limit on how many times per week processed meat can be served and clearer restrictions around maximum sauce quantities, particularly high-sodium sauces such as soy sauce.

Our analysis comparing nutrient provision of meals from internal providers with meals from external providers found that internal meals were significantly higher in a range of nutrients (Table 5). Internal meals were larger (grams) and higher in nutrients, without being higher in energy, indicating that they were more nutrient dense than the external meals. These findings highlight the internal model as being more nutritionally beneficial for students on the programme. Future work should be conducted by Ka Ora, Ka Ako to enable schools to operate an internal model.

The findings from this study add to the body of literature assessing nutrient provision of meals on school lunch programs. They also serve as an indicator of nutritional areas for improvement that can be made on the program. A major strength of this study is that the findings have been directly translated into actionable recommendations and these recommendations have been considered during the development of the new nutrition standards for Ka Ora, Ka Ako. The new nutrition standards are scheduled to be released in Term 4, 2022, and implemented from Term 1, 2023. Another strength of this study is the number of nutrients evaluated. In our study, we analyzed 21 nutrients, whereas many other studies reported only energy and macronutrients, with a select few also reporting sodium, calcium, iron, vitamin A, vitamin C, and fiber.

There are some methodological considerations of this study that should be highlighted. Firstly, the analysis conducted was on meals from approved menus on the program. Therefore, these findings are a measure of expected provision, not actual provision, or intake. This is particularly important to consider when addressing nutrients that were found to be very high (above 100% of the daily requirements) or low (below 30% of RDIs). It is imperative to remember that these are only an accurate indicator of intake if the entire meal is consumed. It is not realistic to assume that all students will consume 100% of the provided lunch meal. In other instances, some schools let students go back for additional serves, if extra food is available. The only way to get an accurate picture of what is being consumed is to collect intake data. Analyzing intake data was outside the scope of this evaluation due to time constraints but should be prioritized in future research and evaluation of the program. In addition, studies should be conducted to assess how students’ nutrient intake from the lunch meal contributes to their daily nutrient intakes. It is expected that children facing food insecurity are unlikely to be meeting the ‘gold standard’ NRV RDIs for nutrients. Therefore, the contribution of nutrients consumed in the school lunch meal compared to their daily nutrient intake could be substantially higher than the % of RDIs the meal is providing. In some instances, the free school lunch meal may be the only meal a child receives that day.

## 5. Conclusions

This was the first nutrition-focused evaluation of the government funded school lunch program in New Zealand. Overall, 77.5% of the 18 analyzed nutrients were above 30% of the RDIs (AI for fiber) and 66.0% were above 1/3 of children’s daily needs. Sodium levels were high compared to international standards and carbohydrate levels (grams) were low compared to international standards. Protein and niacin were provided in levels which exceeded 100% of the daily needs of students in some age groups. Four nutrients were consistently below 30% of the RDIs: energy, iron, calcium, and iodine. The findings from this study have been used to inform the development of revised nutrition standards on Ka Ora, Ka Ako. The revised standards are due for release in the last quarter of 2022. A follow-up study should be conducted to investigate whether the changes in the nutrition standards have resulted in improvements to the nutrient provision on the program, particularly in regard to energy, carbohydrate, and sodium, which could be altered without significant changes to program investment. This study was also conducted on meals from approved menus and therefore reflects expected provision. Future studies should investigate school lunch meal intakes and how they contribute to the daily nutrient intake of students on the program.

## Figures and Tables

**Table 1 nutrients-14-05087-t001:** Meal requirements on Ka Ora, Ka Ako|Healthy School Lunch Program.

Year Level	Minimum Meal Weight	Minimum Protein-Rich Foods	Minimum Vegetable Content
Years 0–3	180 g	30 g animal protein (meat, eggs, fish, cheese etc) OR 60 g plant protein	40 g
Years 4–8	240 g	40 g animal protein (meat, eggs, fish, cheese etc) OR 75 g plant protein	40 g
Years 9+	300 g	50 g animal protein (meat, eggs, fish, cheese etc) OR 90 g plant protein	40 g

**Table 2 nutrients-14-05087-t002:** Breakdown of the meals included in this study.

Year Level	Total Meals	Internal Meals	External Meals
0–3	100	19	81
4–8	97	17	80
9+	105	26	79
Totals	302	62	240

**Table 3 nutrients-14-05087-t003:** Nutrient levels in Ka Ora, Ka Ako meals in each year group category.

	Years 0–3	Years 4–8	Years 9+
Meal Size (g)	241 (200, 322)	300 (264, 339)	378 (332, 470)
Energy (kJ)	1521 (1310, 1795.5)	1773 (1525, 2124)	2332 (1912, 2655)
Protein (g)	20.4 (16.0, 23.9)	22.2 (18.4, 26.9)	31.9 (25.6, 39.2)
Total fat (g)	12.6 (8.2, 15.8)	13.0 (8.9, 17.9)	17.3 (13.3, 21.8)
Saturated fat (g)	4.3 (3.0, 6.9)	4.8 (3.1, 7.6)	6.3 (3.9, 8.9)
Carbohydrates (g)	39.6 (31.1, 49.6)	49.8 (39.6, 59.4)	58.3 (48.8, 75.0)
Fiber (g)	6.4 (4.8, 7.9)	7.7 (6.3, 9.1)	9.0 (7.3, 11.2)
Thiamin (mg)	0.27 (0.21, 0.38)	0.38 (0.21, 0.50)	0.40 (0.31, 0.54)
Riboflavin (mg)	0.23 (0.18, 0.33)	0.27 (0.20, 0.36)	0.35 (0.24, 0.46)
Niacin (mg)	8.1 (6.5, 10.9)	9.1 (6.8, 11.8)	13.3 (10.4, 17.1)
Vitamin C (mg)	14.9 (8.3, 25.4)	16.6 (9.2, 29.6)	23.5 (13.5, 42.2)
Vitamin B6 (mg)	0.28 (0.21, 0.41)	0.31 (0.23, 0.42)	0.40 (0.28, 0.52)
Vitamin B12 (µg)	0.56 (0.27, 1.19)	0.74 (0.38, 1.43)	0.90 (0.38, 1.71)
Folate (µg)	117.9 (70.3, 189.8)	183.8 (88.3, 285.8)	177.2 (99.8, 290.4)
Vitamin A equivalents (µg)	324.0 (179.1, 431.6)	337.4 (189.3, 529.8)	374.9 (227.0, 563.7)
Sodium (mg)	501.8 (346.8, 749.7)	595.3 (451.6, 763.7)	825.1 (535.4, 1090.4)
Magnesium (mg)	63.9 (52.0, 77.2)	78.1 (66.7, 91.2)	99.9 (84.4, 128.7)
Calcium (mg)	181.0 (92.0, 258.2)	216.0 (116.1, 284.6)	257.8 (133.0, 368.1)
Iron (mg)	2.3 (1.7, 3.0)	2.9 (2.2, 3.6)	3.3 (2.7, 4.2)
Zinc (mg)	2.1 (1.6, 3.1)	2.5 (2.0, 3.6)	3.2 (2.5, 4.3)
Selenium (µg)	12.9 (9.6, 17.9)	15.6 (11.3, 20.9)	22.5 (15.2, 31.6)
Iodine (µg)	21.0 (7.9, 34.4)	29.9 (10.5, 47.2)	22.9 (11.3, 56.0)

All reported as median (25th percentile, 75th percentile).

**Table 4 nutrients-14-05087-t004:** Median nutrient levels as a percentage of RDIs.

	Years 0–3 Meals				Years 4–8 Meals				Years 9+ Meals			
Nutrients (% RDI)	5-Year-Olds		8-Year-Olds		9-Year-Olds		12-Year-Olds		13-Year-Olds		18-Year-Olds	
	Female	Male	Female	Male	Female	Male	Female	Male	Female	Male	Female	Male
Energy (kJ)	23.4	21.7	19.8	18.6	21.6	20.1	18.7	16.9	23.3	20.8	21.4	16.7
Protein	**102.1**	**102.1**	**102.1**	**102.1**	63.3	55.4	63.3	55.4	91.2	79.8	70.9	49.1
Thiamin	45.0	45.0	45.0	45.0	42.2	42.2	42.2	42.2	44.4	44.4	36.4	33.3
Fiber ^1^	35.8	35.8	35.8	35.8	38.4	32.0	38.4	32.0	45.1	37.6	41.0	32.2
Riboflavin	38.3	38.3	38.3	38.3	30.0	30.0	30.0	30.0	38.9	38.9	31.8	26.9
Niacin equivalents	**101.1**	**101.1**	**101.1**	**101.1**	75.6	75.6	75.6	75.6	**110.7**	**110.7**	94.9	83.0
Vitamin C	42.7	42.7	42.7	42.7	40.4	40.4	40.4	40.4	58.7	58.7	58.7	58.7
Vitamin B6	46.7	46.7	46.7	46.7	31.0	31.0	31.0	31.0	40.0	40.0	33.3	30.8
Vitamin B12	46.7	46.7	46.7	46.7	41.1	41.1	41.1	41.1	50.0	50.0	37.5	37.5
Folate—dietary folate equivalents	59.0	59.0	59.0	59.0	61.3	61.3	61.3	61.3	59.1	59.1	44.3	44.3
Vitamin A equivalents	81.0	81.0	81.0	81.0	56.2	56.2	56.2	56.2	62.5	62.5	53.6	41.7
Magnesium	49.1	49.1	49.1	49.1	32.5	32.5	32.5	32.5	41.6	41.6	27.8	24.4
Calcium	25.9	25.1	25.1	25.1	17.5	17.5	17.5	17.5	19.8	19.8	19.8	19.8
Iron	23.0	23.0	23.0	23.0	35.8	35.8	35.8	35.8	41.0	41.0	21.9	29.8
Zinc	51.3	51.3	51.3	51.3	41.0	41.0	41.0	41.0	53.0	53.0	45.4	24.5
Selenium	43.0	42.8	42.8	42.8	31.5	31.5	31.5	31.5	45.0	45.0	37.5	32.1
Iodine	23.3	22.9	22.9	22.9	25.1	25.1	25.1	25.1	19.0	19.0	15.2	15.2

Notes: ^1^ Adequate intake. Green cells represent ≥33% of daily requirements. Red cells represent <30% of daily requirements. Yellow cells represent between 30–32% of daily requirements Adequate intake. Contributions greater than 100% of daily requirements are in bold.

**Table 5 nutrients-14-05087-t005:** Significant differences between meals from internal-provision schools and external providers.

	**Internal**	**External**	***p*-Value**
**Years 0–3**			
Meal size (g)	341 (259, 431)	227 (197, 276)	<0.001
Protein (g)	24.9 (17.5, 30.4)	20.1 (15.5, 22.7)	0.013
Fiber (g)	7.6 (6.1, 10.1)	6.2 (4.8, 7.6)	0.011
Niacin equivalents (mg)	9.8 (8.1, 13.8)	7.6 (6.4, 9.9)	0.008
Vitamin C (mg)	24.0 (14.6, 57.1)	13.9 (7.4, 22.8)	0.004
Vitamin B6 (mg)	0.42 (0.27, 0.52)	0.27 (0.21, 0.37)	0.011
Magnesium (mg)	82.7 (60.6, 125.4)	62.0 (47.4, 73.8)	0.002
Iron (mg)	2.9 (1.9, 4.2)	2.2 (1.7, 2.9)	0.046
**Years 4–8**			
Meal size (g)	316 (280, 430)	292 (262, 332)	0.026
Vitamin C (mg)	34.2 (14.5, 40.8)	15.0 (8.9, 24.5)	0.006
Vitamin B12 (µg)	1.42 (0.89, 2.04)	0.65 (0.36, 1.09)	0.003
**Years 9+**			
Meal size (g)	486 (372, 592)	360 (320, 431)	<0.001
Energy (kJ)	2759 (2310, 3042)	2143 (1835, 2556)	<0.001
Protein (g)	40.9 (31.5, 46.4)	29.8 (24.5, 36.8)	<0.001
Carbohydrates (g)	9.7 (7.9, 12.7)	9.0 (6.9, 10.6)	0.042
Niacin equivalents (mg)	17.1 (11.3, 20.7)	12.6 (9.8, 16.2)	0.014
Magnesium (mg)	120.8 (95.0, 139.2)	96.3 (81.3, 122.0)	0.013
Zinc (mg)	3.8 (3.1, 5.3)	3.0 (2.2, 4.0)	0.007

## Data Availability

The data presented in this study are available on reasonable request from the corresponding author.

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
