# Peer review of "Nutrient-Level Evaluation of Meals Provided on the Government-Funded School Lunch Program in New Zealand"

_nutrients, 2022, doi:10.3390/nu14235087_

Round 1

Reviewer 1 Report

I found the research well structured and presented. As the authors affirm themselves, the originality is guarantee by they proposal of a nutrient-level school lunch program standards in New Zealand, where no other are there, and the study about the relative impact. ("This was the first nutrition-focused evaluation of the government funded school 474 lunch program in New Zealand").

Authors may decide add some considerations to their conclusions.

Author Response

The authors would like to thank the reviewers for their feedback on our manuscript entitled "Nutrient-level evaluation of meals provided on the government-funded school lunch program in New Zealand". Our responses to the reviewers' comments have been listed in italicised text below each of the comments in the attached file and the changes in the manuscript have been highlighted in yellow. We hope these revisions will be to the satisfaction of the reviewers and very much look forward to having this work published in Nutrients.

Reviewer 2 Report

That interesting work about evaluation of the School Lunches ongoing program in New Zealand. This study analyzed the nutritional content of lunch meals and compared them to national and international standards. In my opinion it is not necessary refer to international standards in nutrition-based evaluation the meals served at school? Each country adopts its own Nutrient Reference Values that take into account the results of eating habits and nutritional status of the population living under certain environmental conditions.

In the introduction and discussion, I propose to shorten the text on the importance of nutrients for the proper functioning of the body, as well as cognitive functions.

In part Materials and Methods  it is not necessary to provide a definition of the levels of norms for specific nutrients. But in Line 191 sodium levels in the meals were  compared to Upper Levels (UL) in the NRVs. Why for sodium was UL selected in the evaluation and not an adequate intake (AI)?

Please, check the attached publication on the importance of sodium in human nutrition  https://doi.org/10.2903/j.efsa.2019.5778

In this study especially, comparison of nutrient content of meals for children from internal and external providers is very interesting.

I recommend that you improve the description of the results, because after the presentation of the results from table 4, the results from table 3 are re-described. Please, check the correctness of the energy unit notation in text and table 3.

The discussion requires improvement. For example, in text "The main contributor of sodium on Ka Ora, Ka Ako was wholemeal bread products", but the presented results did not include sources of nutrients?

Author Response

(The authors gave the same response as above.)

Round 2

Reviewer 2 Report

Thank you for your answers and for including selected changes in the text of the work.

However, kJ is a standard in the presentation energy unit, I meant only the issue of the correctness of the writing?

In supplementary materials you provide energy (cals) (rather kcal ?), it is worth unifying the units used.

The discussion lacks reference to the results of the comparison of nutrient content of meals for children from internal and external providers.

Author Response

The authors would like to thank the reviewer for their feedback on our manuscript entitled "Nutrient-level evaluation of meals provided on the government-funded school lunch program in New Zealand". Our responses to the reviewer’s comments have been listed in italicised text below each of the comments and the changes in the manuscript have been highlighted in yellow. We hope these revisions will be to the satisfaction of the reviewer and editor and very much look forward to having this work published in Nutrients.
